



# Matrix gas flow through 'impermeable' rocks - shales and tight sandstone

Ernest Rutter[1], Julian Mecklenburgh[1], Yusuf Bashir[1,a]

[1]Rock Deformation Laboratory, Dept. of Earth and Environmental Sciences, University of Manchester, Manchester M13 9PL, UK.

[a]Now at: Department of Petroleum Resources, Abuja, Nigeria.

*Correspondence to :* E. Rutter (e.rutter@manchester.ac.uk)

**Abstract.** The effective pressure sensitivity of gas flow through two shales (Bowland and Haynesville shales) and a tight gas sandstone (Pennant sandstone) was measured over the typical range of reservoir pressure conditions. These are low permeability rocks such as can be exploited as caprocks above reservoirs that might be developed to store compressed air, methane, hydrogen or to bury waste carbon dioxide, all of which may become important components of the forthcoming major changes in methods of energy generation and storage. Knowledge of the petrophysical properties of such tight rocks will be of great importance in such developments. All three rocks display only a small range in $\log_{10}$ permeability at low pressures, but these decrease at dramatically different rates with increasing effective pressure, and the rate of decrease itself decreases with pressure, as the rocks stiffen. The pressure sensitivity of the bulk moduli of each of these rocks was also measured, and used to formulate a description of the permeability decrease in terms of the progressive closure of narrow, crack-like pores with increasing pressure. In the case of the shales in particular, only a very small proportion of the total porosity takes part in the flow of gases, particularly along the bedding layering.

Key words: Permeability. shales, sandstone, bulk modulus, pressure sensitivity, gas porous flow

Supplementary data file: DF1.csv at https://zenodo.org/record/5675601

## 1. Introduction

Shales (laminated mudstones) are of particular importance because their fine grain size and tight pore structure gives them a particularly low matrix permeability and hence makes them excellent cap rocks for the containment of oil, water and gases. This includes their future use as a sealant for the storage containment of fuel gases hydrogen and methane, compressed air storage and for the disposal deep underground of waste liquids and gases, including waste carbon dioxide. Organic shales are source rocks for petroleum and become source, reservoir and seal for unconventional natural gas (shale gas). The enormous economic importance of shales cannot be overstated, and this demands an ever-increasing understanding of their petrophysical properties.

Compared to conventional reservoir rock materials (sandstones, limestones), shales are particularly difficult to work with. Their commonly laminated nature makes them often highly fissile, with a tendency to split along the layering. Thus coring and cutting operations for sample preparation are often difficult, and their physical properties (elasticity, mechanical strength, permeability, elastic wave velocities) are generally anisotropic. Determination of properties that involve working with elevated pore pressures become time-dependent, according to the slow rates of fluid permeation though the microstructure in response to applied effective pressure changes, and the rock itself may display time-dependent deformation (creep) under load. Mineralogically, shales can be highly variable, particularly with respect to the relative proportions of the major mineral components: framework silicates, clays and other phyllosilicates, and carbonates (Lazar et al. 2015; Diaz et al. 2013; Dowey andTaylor 2020), and this can be expected to be reflected in the spectrum of petrophysical properties of shales.



In contrast to shales, tight gas sandstones (e.g. Zee Ma et al., 2016)) may display similarly low permeabilities
and porosities, but lack extreme fissility and typically possess a matrix of coarser-grained framework silicate
minerals (quartz and feldspar) but with primary pore spaces filled with some detrital micas but also authigenic
growths of clay minerals and hydrated oxide phases. Thus their properties tend to form an upper (more permeable
and less anisotropic) bound to the range of properties displayed by shales. For this reason, we have included for
comparison in this study such a rock type.  Here also we present a study of the matrix permeability of two, rather
different shales. Permeability and storativity were measured parallel to the layering under hydrostatic loading
conditions as a function of total confining pressure and pore pressure of argon gas, and normal to layering at one
pore pressure only. Results were fitted to a simple physical model. The spectrum of behaviours observed provides
insight into the physical controls on the matrix permeabilities of these rocks.
**2.    Sample materials and characterization**
Two shale samples recovered from depth in boreholes were used. The samples are strikingly mineralogically
and microstructurally different. They were characterized mineralogically by quantified X-ray diffraction analysis,
which was also used to estimate grain density using published mineral densities. All samples were oven dried at
60 ºC until constant weight (at least one week), and then maintained at that temperature until use. All experiments
were carried out in this oven dried state. Other than with the degree of water saturation in the as-supplied state, it
can be very difficult to test shales with varying degrees of controlled or with total water saturation. The sandstone
studied was from a surface exposure but was treated in the same way as the shales.
**2.1 Pennant sandstone.**
This is a hard, grey marine sandstone (Fig. 1a and b) of upper Carboniferous age (Kelling 2017), that outcrops
in south Wales, Great Britain. We have previously reported rock mechanics studies on this rock in Hackston and
Rutter (2016) and Rutter and Hackston (2017). All measurements reported were made normal to bedding. Bedding
planes are not apparent in hand specimen.
Modal proportions (vol% solids):   Quartz + Feldspar 73.73 ;  Phyllosilicates  9.81 ; (estimated uncertainties  ±
4% of cited percentages)
Grain density 2661 ± 120 kg/m$^3$ :  Bulk density 2558 ± 35  kg/m$^3$: Total porosity 3.89 ± 0.04 % from XRD,
4.60% ± 0.01 using a helium porosimeter.
**2.2 Bowland Shale.**  This is a phyllosilicate-rich, carbonate-poor siliceous mudstone (Fig. 1c), very pyrite-rich,
(8.3 wt%), of Lower Carboniferous age. It was the target formation for exploitation of shale gas in Northern
England.
Depth 2060.55 m. Provider sample identifier IG 5-8W. Location: west Manchester, UK.
Modal proportions (vol% solids):   Quartz + Pyrite 38.4 ;  Phyllosilicates  61.6 ; Carbonates  0 (estimated
uncertainties  ± 4% of cited percentages)
Grain density 2842 ± 120 kg/m$^3$ :  Bulk density 2714 ± 38 kg/m$^3$: Total porosity 4.50 ± 0.02% from XRD; 4.6% ± 0.1
using a helium porosimeter.
Total organic carbon 1.14 ± 0.2 wt% ;  Water loss from drying 0.74 ± 0.15 vol% , hence initial water saturation =
76  13%.

**2.3 Haynesville Shale**



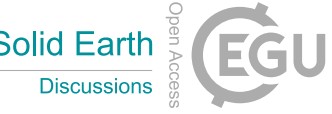

This is a phyllosilicate-poor, carbonate-rich siliceous mudstone (Fig. 1d). Pyrite-poor (0.7 wt%), of Upper
Jurassic age (Hammes et al. 2011), successfully exploited for shale gas in the southern United States.
Depth 3730.6 m. (Sample identifier). Location: Hewitt Land LLC well, Caspian Field, de Soto parish, Louisiana,
USA.
Modal proportions (vol% solids):   Quartz + Feldspar + Pyrite 34.5;   Phyllosilicates 13.4;   Carbonates 52.1;
(estimated uncertainties ± 4% of cited percentages)
Grain density $2703 \pm 120$ kg/m$^3$:  Bulk density $2453 \pm 35$ kg/m$^3$: Total porosity $9.26 \pm 0.04$ % from XRD, $7.6\% \pm 0.1$
using a helium porosimeter. Total organic carbon $1.3 \pm 0.2$ wt%.

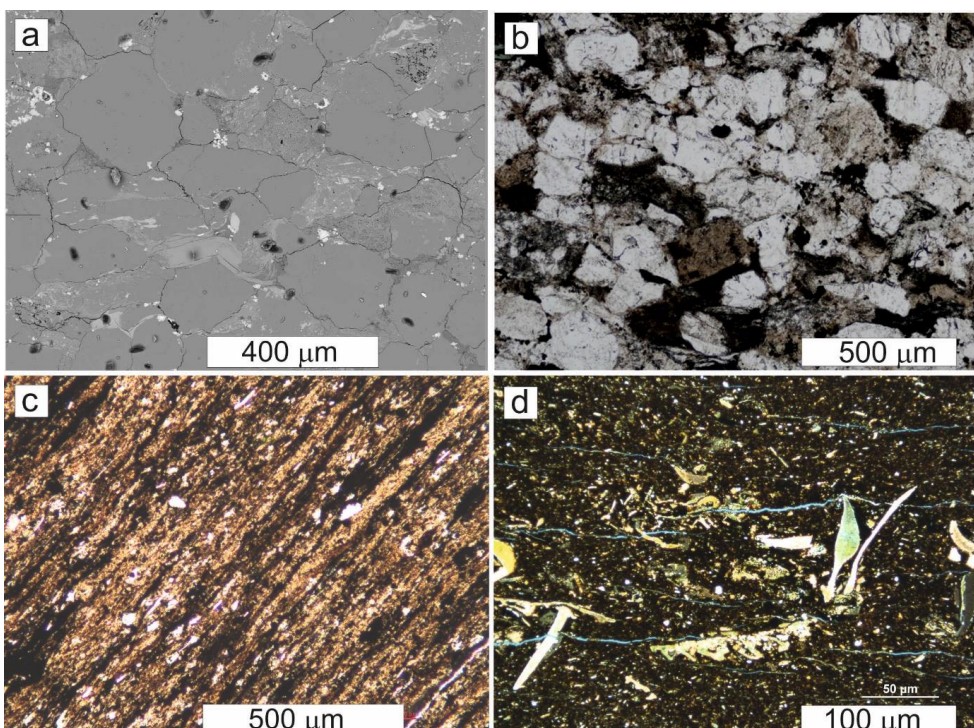


**Figure 1. Microstructures of the rocks tested.**
**(a) Back-scattered electron (BSE) image and (b) optical image (PPL) of Pennant sandstone, bedding**
**horizontal, showing large quartz grains (mid-grey in (a)) with sutured contacts caused by pressure solution**
**and remaining pore spaces largely filled by iron hydroxide (white in (a)) and authigenic clay minerals (light**
**grey in (a)), reducing the overall porosity to 4.6%**
**(c) Microstructure (Plane-polarized light (PPL) image of polished thin section) of Bowland shale, finely and**
**homogeneously banded with elongate clusters of organic material and pyrite (black) and silt-sized grains of**
**quartz in a matrix of elongate clusters of phyllosilicate (clay + detrital micas) grains.**
**(d)  Microstructure of Haynesville shale. (PPL image of polished thin section, horizontal dimension is**
**parallel to layering).  Bioturbation destroys continuity of layering. Rock is only weakly banded but**
**nevertheless fissile; bedding-parallel cracks can be seen, opened during thin-section preparation.**
**Calcareous fossil fragments and authigenic calcite-filled voids, in matrix of finer grained phyllosilicate**
**(clays + detrital micas) and fine silt-sized framework silicates.**





Defining velocity anisotropy as $2(V_{max} - V_{min})/(V_{max} + V_{min})$, the anisotropies of Bowland and Haynesville shales
are respectively  30.7% and 32.2% at 100 MPa total confining pressure. The velocity anisotropy of Pennant
sandstone at elevated pressure was not determined. It is 15.5% axial and 3.1% radial at room pressure, but will be
less at elevated pressure.
The wt% values for the mineralogical composition of all rock types were converted to vol% using tabulated
densities from the literature (Mavko et al., 2009; Mondol et al., 2008), and together with averaged mineral elastic
properties the bulk elastic properties of the rocks estimated were as Voigt-Reuss-Hill (VRH) averages assuming
zero porosity. These are listed in Table 1.
Some comparisons of behaviour are made with previously published (Mckernan et al., 2017) data on Whitby
shale. This is a well-foliated, silt-bearing, clay-rich, carbonate-poor mudstone of Liassic age, with 8.1% total
porosity and 1.5% volume amorphous organic matter.


**Table 1.  Phase fractions, mineral densities and Voigt-averaged bulk and shear moduli  $K_v$ and $G_v$  (from**
**literature) and calculated zero porosity elastic moduli as Voigt-Reuss-Hill (VRH) averages (GPa) for**
**Bowland and Haynesville shales and for Pennant sandstone. Organic fraction not included. Mineral phase**
**Reuss-average elastic moduli can be calculated from the other values supplied. $K_0$ = bulk modulus, $G_0$ =**
**shear modulus, $E_0$ = Young's modulus (VRH-averaged whole-rock values assuming isotropy). Modal**
**volume percent is % of the solids.**

| Bowland Shale IG5-8WC | | | | | | |
|---|---|---|---|---|---|---|
| Phase | Wt% | ±Error% | Density kg m$^{-3}$ | Vol% | Kv GPa | Gv GPa |
| Quartz | 30.98 | 1.42 | 2648 | 33.64 | 12.73 | 14.90 |
| Pyrite | 8.32 | 0.44 | 5020 | 4.77 | 6.63 | 5.36 |
| Muscovite 2M | 60.44 | 2.04 | 2844 | 61.11 | 35.55 | 21.61 |
| Kaolinite | 0.26 | 2.60 | 1580 | 0.48 | .0072 | .0067 |
| Total | 100.0 | | | 100.0 | | |
| Zero porosity moduli (GPa): | VRH($K_0$) | | VRH($G_0$) | | VRH($E_0$) | |
| | 52.79 | | 40.69 | | 97.13 | |
| | | | | | | |


| Haynesville Shale YB03 | | | | | | |
|---|---|---|---|---|---|---|
| Phase | Wt% | ±Error% | Density kg m$^{-3}$ | Vol% | Kv GPa | Gv GPa |
| Albite | 10.49 | 0.505 | 2610 | 11.01 | 5.59 | 3.22 |
| Ankerite Fe0.55 | 4.65 | 0.36 | 3050 | 4.17 | 4.80 | 2.46 |
| Calcite | 47.22 | 1.25 | 2712 | 47.69 | 32.94 | 15.24 |
| Clinochlore IIb-24.11 | 0.41 | 2.90 | 3880 | 2.26 | 1.37 | |
| Muscovite 1M | 9.97 | 1.46 | 2844 | 9.50 | 5.27 | 3.36 |
| Pyrite | 1.27 | 0.10 | 5020 | 0.69 | .958 | .775 |
| Quartz | 18.71 | 0.74 | 2648 | 19.35 | 7.32 | 8.57 |
| Siderite | 0.47 | 0.07 | 3960 | 0.33 | .408 | .168 |
| Orthoclase | 3.20 | 0.46 | 2540 | 3.45 | 1.61 | .815 |
| Total | 100 | | | 100.1 | | |
| Zero porosity moduli (GPa) | | | | VRH($G_0$) | VRH($E_0$) | VRH($K_0$) |
| | | | | 60.57 | 34.91 | 87.86 |





| Pennant Sandstone Pe2 | | | | | | |
|---|---|---|---|---|---|---|
| Phase | Wt% | ±Error% | Density kg m$^{-3}$ | Vol% | Kv GPa | Gv GPa |
| Albite | 16.14 | 0.70 | 2610 | 16.46 | 8.20 | 4.72 |
| Phyllosilicates | 10.48 | 1.5 | 2840 | 9.81 | 6.10 | 3.71 |
| Quartz | 73.37 | 2.8 | 26480 | 73.73 | 27.77 | 32.50 |
| Total | 99.99 | | | 100.0 | | |
| Zero porosity moduli (GPa): | | | | VRH(K$_o$) | VRH(G$_o$) | VRH(E$_o$) |
| | | | | 41.55 | 40.42 | 91.57 |


------------------------------------------------------------------------------------------------------------

**3.    Experimental Methods**
**3.1  Permeability measurements**
Permeability measurements were made on cylindrical samples of either 25.4 or 20 mm nominal diameter, cut to
lengths of the same order or shorter. The latter is generally necessary for very low permeability rocks, but quite
apart from this it was not possible to obtain long cores from slabbed drill cores of the shales. Problems were also
encountered during shale specimen preparation owing to the friable nature of these materials.  Porous sintered
stainless steel (316L) filter plates (17% porosity) were placed at either end of the sample to spread the pore fluid
uniformly over the ends of the rock samples. The assembly was jacketed in a heat-shrinkable polymer jacket, so
that pore fluid pressures less than the confining pressure could be applied. Confining pressures (hydraulic oil, a
synthetic ester, di-octyl sebacate, trade name Reolube DOS®) ranging up to a little over 100 MPa were used. This
fluid has the advantage of a relatively small rate of change of viscosity with pressure (see Rutter and
Mecklenburgh 2017 and 2018 for further details). In all experiments argon gas was used as the pore fluid, at
pressures ranging up to 80 MPa. The higher viscosity of a liquid pore fluid would have led to very long
experimental durations. The confining and pore pressures ranges cover the full extent of likely pressures to be
encountered in engineering operations to depths of *ca* 4 km.

Although it was intended that experiments would be carried out under hydrostatic confinement conditions, the

presence of a contrast in elastic properties of the specimen against the porous end plates and the steel loading
pistons induces a shear stress along these interfaces. This in turn causes the stress state in the specimen to deviate
from hydrostatic and to reduce the average mean stress. Deviations from hydrostatic loading are most severe when
the length of the specimen becomes less than twice the diameter. For this reason, mechanical testing of rocks is
usually carried out on specimens with a length:diameter ratio of 2.5:1 or more. Finite element analysis (FEA) of
the stress state in rocks confined between steel end plates were carried out to assess the expected departures from
hydrostatic loading, and the effects predicted must be borne in mind when interpreting the permeability data.
**Table 2. Elastic constants of the components in the finite element models.**

| | Young's Modulus $E$ GPa | Poisson's Ratio |
|---|---|---|
| Sample | 60 | 0.250 |
| Piston | 190 | 0.265 |
| Spacer (17% Porosity) | 108.6 | 0.260 |







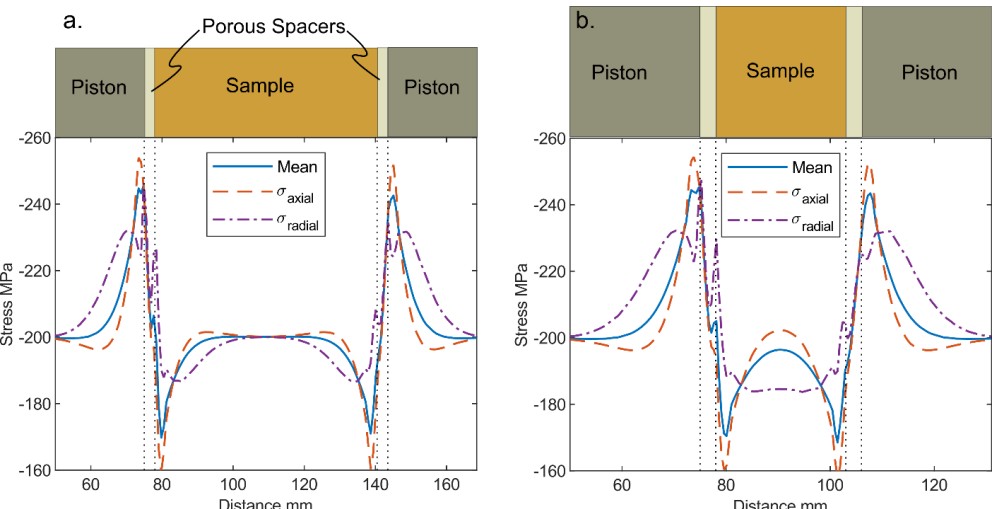


**Figure 2. Results of finite element analyses showing stress profiles of mean stress, axial normal stress and**
**radial normal stress along the axes of samples respectively of length:diameter ratios (a) 2.5:1 and (b) 1:1,**
**each with a diameter of 25.4 mm. At the top of each figure is a scaled schematic of the assembly; notice the**
**aspect ratio of the sample in each case. Externally applied hydrostatic stress was 200 MPa. For the longer**
**sample the stress state in the greater part of the sample was near homogeneous, but for the shorter one a**
**differential stress on the order of 7% of the applied hydrostatic stress was induced.**

   Figure 2 presents the results of finite element analyses showing stress profiles along the axes of samples
respectively of length:diameter ratios (a) 2.5:1 and (b) 1:1, with a hydrostatic pressure of 200 MPa applied to the
outer cylindrical surfaces. At each end of the sample a 3 mm thick, porous sintered steel disk was placed.
Positions of boundaries between the solid steel pistons, the porous disks and the sample material are indicated. In
both cases the sample diameter was 25.0 mm. Along-axis stress component variations were more varied than
across the radius. Most of the stress heterogeneity (departure from the applied 200 MPa hydrostatic pressure)
resides in and immediately adjacent to these disks, and for each stress component is of similar magnitude for both
specimen lengths. Within the greater part of the sample volume in each case the axial normal stress is higher than
the radial normal stress, and these components are similar to the principal stress values. For the longer sample, the
stress state is near hydrostatic over 0.8 of the specimen length, but in the case of the shorter sample the stress
components are notably non-hydrostatic over most of the specimen length, with maximum differential stress
reaching 15 MPa (7% of the applied hydrostatic stress) in the central part of the sample.

   A small number of permeability measurements were made using the pulse-transient-decay method of Brace et al.
(1968), as modified by Cui et al. (2009). However, most experimental results were obtained using the oscillating
pore pressure method (Kranz et al., 1990; Fischer and Paterson, 1992;  Faulkner and Rutter, 2000; Bernabé et al.,
2006; Mckernan et al., 2017). Whilst keeping the confining pressure constant and after establishing a constant
pore pressure in the sample, a sinusoidal oscillation of pore pressure, of known period and of amplitude about 1
MPa, is applied at one end of the sample (upstream). As the pressure wave propagates through the sample it is
phase-shifted and loses amplitude. The amplitude ratio (gain) and phase shift angle are measured. The solution to





the transport equation for these measured parameters is given by Bernabé et al., (2006) in terms of two

dimensionless numbers, $\eta$ and $\xi$, from which permeability and sample storativity can be calculated using

$$\xi = \frac{SL\beta}{\beta_D}, \quad \eta = \frac{STk}{\pi L \mu \beta_D} \qquad (1)$$

Here, $S$ is cross sectional area of the sample (normal to flow path), $L$ is specimen length, $\beta_D$ is downstream
volume storativity and $\beta$ is specimen storativity, $T$ is the period of the pore pressure oscillation, $k$ is specimen
permeability, and $\mu$ is viscosity of the pore fluid. Argon gas viscosity as a function of pressure data was reported
by Michels et al., (1954). Storativity is the product of the volume of the void space concerned, with the pore fluid
(isothermal) gas compressibility. Argon compressibility is non-linear over the pore pressure range used (Gosman
et al., 1969) and substantially non-ideal above about 20 MPa. $\xi \approx \phi V_s / V_d$ where $\phi$ is specimen effective porosity,
$V_s$ is total specimen volume and $V_d$ is downstream reservoir volume. It cannot be assumed that effective
(conductive) porosity estimated from permeability measurements will necessarily be equal to total porosity
measured independently.
The apparatus used was the same as used for experiments reported by Rutter and Mecklenburgh (2017; 2018).
Pressure transducers with a resolution of 0.02 MPa were used for pore pressure measurements, and confining
pressure was measured to an accuracy better than 0.3 MPa. The minimum pore pressure used was 10.0 MPa. This
is sufficiently high to avoid exsorption of gas from mineral surfaces and to avoid slip flow of gas through pore
spaces (Knudsen/Klinkenberg effect, Mckernan et al. 2017). It was determined that the experimental assembly
shows no detectable gas flow when a rock sample is replaced by an impermeable steel plug.
**3.2 Error, uncertainty and reproducibility**
Accuracy of reported permeability depends on uncertainties of the parameters in Eq. (1). $\eta$ and $\xi$ can be
measured to within about 2% of the true value, and $S, T L$ and $\mu$ to within 1%. The least certainly known
parameter is the downstream volume, which is determined as the difference between the total volume of the pore
pressure pipework measured with and without the downstream pipework connected, each measured by the pore
pressure change produced by a known volumometer piston displacement. The downstream reservoir volume $V_d$
was measured to be $445 \pm 30$ mm$^3$, including the volume of the downstream porous steel filter. These
uncertainties translate to an accuracy of $\log_{10}$ permeability of $\pm 0.1$ log units. This is small, given that permeability
varies with pressure by 1 to 3 orders of magnitude.
The largest apparent uncertainties in reported permeability data arise from hysteretic changes in the behaviour of
the rock itself as effective pressure is cycled and will be discussed when the data are presented.
**3.3 Bulk modulus measurements**
Bulk modulus measurements as a function of confining and pore pressures were made as far as possible on
physically the same samples that were used for the permeability measurements, to avoid any influence of
mineralogical or microstructural differences. Measurements were made over a range of total confining pressures
up to 200 MPa, after the permeability measurements were made, with constant pore pressures of argon gas,
typically at nominally 10, 35, 67 and 100 MPa. The method involved measuring volume of pore fluid (argon gas)
progressively expelled as the total confining pressure was increased at constant pore pressure. This measures the





compressibility of the pore spaces. P-wave acoustic velocity measurements were made at the same time, although
these data are not reported here.
Unlike for permeability measurements, porous steel plates were not used at the ends of the specimens for pore
fluid displacement measurements. For the relatively porous and permeable Haynesville shale and Pennant
sandstone, a short hole, normally 15 mm long and 1.5 mm diameter, was drilled into the end of the specimen
facing the pore pressure inlet pipe, to facilitate flow of gas into and out of the specimen. This was thought to be
unlikely to be adequate for the lower porosity and permeability Bowland shale, therefore samples were cut in half
parallel to the long axis so that a 2mm thick, porous steel plate could be inserted, to facilitate gas flow over a wide
surface area of the rock, yet without affecting the P-wave velocity along the length of the specimen.
When considering the results, the procedure for pressure application is of importance. For the tests with pore
pressure, the application of a confining pressure slightly greater than the eventual pore pressure was made,
followed by application of the pore pressure. Then the total confining pressure was increased stepwise away from
the constant pore pressure. Thus tests at high pore pressure have been exposed to much higher effective pressures
before application of pore pressure, than when the test pore pressure is to be low.
When pore pressure was made non-zero, constant pore pressure was maintained using a servo-controlled pore
volumometer.  Each applied increment of the confining pressure caused a small elastic contraction of the pore
volume that attempts to raise the pore pressure. The servo-controller backs off the moveable piston in the pore
volumometer in order to keep the pore pressure constant. The distance swept by the volumometer piston at
constant pore pressure allows the volume of gas expelled to be measured to a resolution of 0.4 mm$^3$. In this way
the history of *pore volume* change at constant pore pressure during progressive loading by the confining pressure
can be determined. The compressibility of the pore space $C_{pc}$ is given by the fractional change in pore volume $V_p$
in response to a change in confining pressure $P_c$ at constant pore pressure $P_p$ (Zimmerman, 1991), and is the
reciprocal of the dry pore space bulk modulus $K_\phi$ :
$$C_{pc} = \frac{1}{K_\phi} = \frac{1}{V_p}\left(\frac{\partial V_p}{\partial P_c}\right)_{Pp} \qquad (2)$$

Note $V_p = \phi V_b$ , where $V_b$ is the total sample volume.  $K_{dry}$ is the bulk modulus of the porous aggregate. Its
reciprocal, compressibility $C_{bc}$ , the bulk volume change in response to a change in confining pressure at constant
pore pressure, is defined by
$$C_{bc} = \frac{1}{K_{dry}} = \frac{1}{V_b}\left(\frac{\partial V_b}{\partial P_c}\right)_{Pp} \qquad (3)$$

where $V_b$ is the bulk volume, including the pore space. The zero-porosity bulk modulus of the constituent mineral
aggregate is defined as $K_o$ (Table 1), then the dry bulk modulus $K_{dry}$ (= $K_{bc}$) is given (Mavko et al., 2009) by
$$\frac{1}{K_{dry}} = \frac{1}{K_o} + \frac{\phi}{K_\phi} \qquad (4)$$

Decrease in permeability with increasing Terzaghi effective pressure ($P_c - P_p$)  (Terzaghi, 1923) is primarily due to
the pressure dependence of $K_{dry}$, leading to progressive closure of pore space. Thus the independent determination of
$K_{dry}$ from pore volumometry measurements provides a basis for the interpretation of the pressure sensitivity of
permeability.





Note that we have no means of measuring directly the influence of pore pressure change on bulk deformation of
the sample, characterized by the compressibility $C_{bp}$, or
$$C_{bp} = \frac{1}{K_{bp}} = \frac{1}{V_b}\left(\frac{\partial V_b}{\partial P_p}\right)_{Pc} \qquad (5)$$

This would require strain gauges or equivalent to be mounted on the outer surface of the rock sample (e.g.
Hasanov et al., 2019, 2020). However, it can be obtained from
$$\frac{1}{K_{bp}} = \frac{1}{K_{bc}} - \frac{1}{K_o} \qquad (6)$$

(Mavko et al., 2009).
Biot and Willis (1957), Skempton (1960) and Nur and Byerlee (1971) obtained a theoretical expression for the
effective pressure coefficient (Biot coefficient) *m* for elastic *deformations* (including deformations of pore spaces)
of a mechanically linear, homogeneous and isotropic rock, so that effective pressure $P_{eff} = (Pc - mP_p)$, and
$$m = 1 - \frac{K_{dry}}{K_o} \qquad (7)$$

Note that this effective pressure coefficient is not necessarily the same as that describing empirically the influence
of pore pressure on permeability, nor on elastic wave velocities nor the failure characteristics of rocks (whether
frictional sliding or intact rock failure).
*m* is also given by
$$m = \frac{K_o}{K_{bp} + K_o} = \frac{K_o - K_{bc}}{K_o} = 1 - \frac{K_\phi}{K_\phi + K_0\phi} \qquad (8)$$

Sample storativity is related to these stiffness parameters by
$$\beta = \frac{1}{K_{bp}} + \phi\left(\frac{1}{K_f} - \frac{1}{K_o}\right) \qquad (9)$$

where $K_f$ is pore fluid bulk modulus (Hasanov et al., 2019).
In all calculations we assume $K_o$ is negligibly sensitive to effective pressure, compared to porous rock stiffnesses
such as $K_{dry}$, following data for $K_o$ for minerals such as quartz via ultrasonic measurements (e.g. Calderón et al.,
2007, who give $K_o = 37.5(GPa) + 4.7*P(GPa)$).
**4. Experimental results**
A full tabulation of experimental results is given in the supplementary data file DF1.
**4.1 Permeability results**
Experimental conditions and results are presented graphically in Figs. 3 through 8. The first pressure cycle
applied to most rocks results in higher permeabilities and a relatively rapid rate of decrease of permeability with
pressure, as inelastic cracks become progressively and permanently closed. Subsequent pressure cycles up to the
maximum pressure previously attained are more nearly elastic and reproducible, although there can be a small
tendency to reduce permeability slightly with subsequent pressure cycles. The first stage in a suite of permeability
measurements covering a wide range of confining and pore pressures therefore must be to take the sample to the





maximum effective pressure to which it is to be exposed, to ensure closure of these inelastic cracks and pores up
to that pressure.

### 4.1.1 Form of data and reproducibility

In the regime of elastic behaviour permeability (as log $k$) is not usually linear neither on a $k$ vs $P_c$ plot nor even
on a log $k$ vs $P_c$ plot but is concave upwards (Fig. 3). The decrease of permeability with effective pressure is due
to elastic closure of conductive cracks and pores, and this is expected to become more difficult as the porous
material stiffens at higher pressure. Thus although it is common, and useful for the purpose of modelling reservoir
behaviour (e.g. Kwon et al., 2001; Bustin et al., 2008; Cui et al., 2009; Heller et al., 2014; Mckernan et al., 2017)
to describe quantitatively the relationship between $\log_{10} k$ and $P_c$ by making a least-squares linear fit to the data, a
better description would take into account the curvature.
In order to estimate the reproducibility of the permeability data, a determination of the standard error was made
about a polynomial fit to the 10 MPa pore pressure data (after the first pressure cycle) for each rock type. For
Bowland shale it is ±0.22 $\log_{10} k$ units, for Haynesville shale it is ±0.19 $\log_{10} k$ units and for Pennant sandstone it is
±0.10 $\log_{10} k$ units.

### 4.1.2 Influence of confining (Pc) and pore pressures (Pp) on permeability

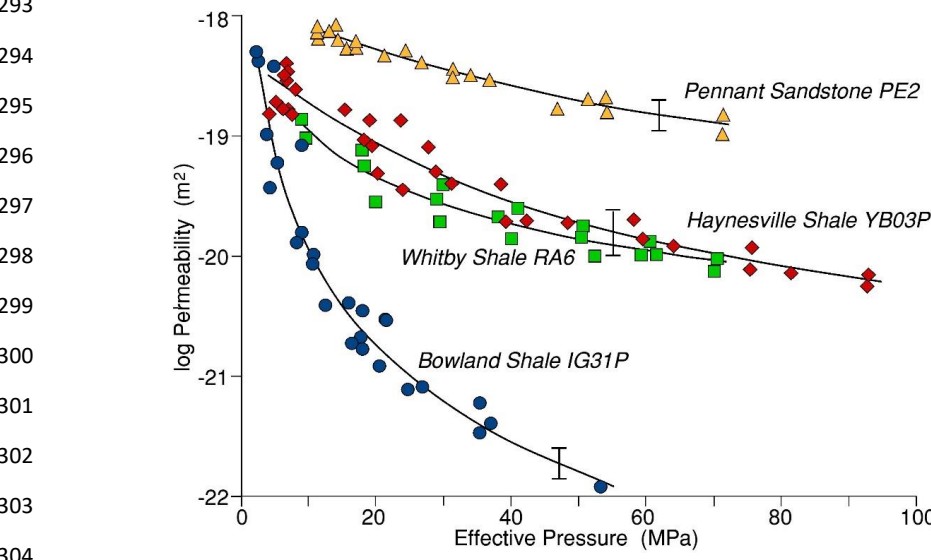

**Figure 3. Matrix permeability of Pennant sandstone for flow normal to bedding, and for Bowland and Haynesville shales for flow parallel to layering, as a function of Terzaghi effective pressure ($P_c – nP_p$) over a wide range of pore pressures of argon gas. Data of Mckernan et al. (2017) for Whitby shale sample RA6 at a constant argon gas pore pressure of 25 MPa are also shown for comparison. In each case data from the first pressure cycle up to the maximum effective pressure attained has been excluded. All rocks show permeability decreasing more slowly with effective pressure at higher effective pressures. Error bars are shown as estimated for the 10 MPa pore pressure data.**

Figure 3 shows the influence of Terzaghi effective pressure on matrix permeability over a range of pore
pressures, for Haynesville and Bowland shales for flow parallel to layering and for Pennant sandstone normal to
bedding after the first pressure cycle. They are expressed as $\log_{10} k$ versus effective pressure ($P_c – nP_p$), where



the pore pressure parameter $n = 0.86$ for Pennant sandstone and is 0.99 for Haynesville shale. For the Bowland
shale the data showed that permeability varied over almost four orders of magnitude, much greater than for the
other two rock types, and as a result it was evident that parameter $n$ tended to increase with the value of Terzaghi
effective pressure, varying from unity at low pressures to 1.6 at high effective pressures. The least squares best-
fit curve to each of these data sets is shown in Fig. 3. For all three rocks the form of the behaviour is similar, each
showing a decreasing slope at higher effective pressures, as would be expected from pressure-induced
constriction of pore spaces. The Pennant sandstone showed the least sensitivity to effective pressure variations,
whilst the Bowland shale displays a far greater sensitivity to effective pressure. The Haynesville shale takes an
intermediate position that is closely comparable to the data for Whitby shale (sample RA6 taken from the data
reported by Mckernan et al. 2017 for pressure cycles 2, 3. 4 and 5).
Whilst these rocks display relatively small differences in permeability at low effective pressures, increase in
pressure results in markedly divergent trends, resulting in large differences in permeability developing over the
range of effective pressures expected to encountered under reservoir conditions. This observation emphasises the
importance of understanding the pressure sensitivity of shales that are to be exploited for engineering purposes.
**4.1.3 Influence of flow direction at constant pore pressure.**

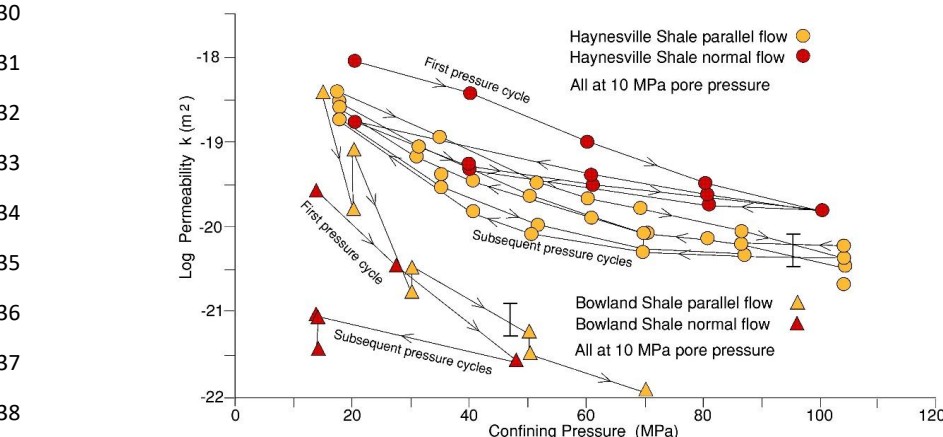

**Figure 4. Comparison of data at 10 MPa pore pressure for flow parallel and normal to layering in the two**
**shales. Parallel flow data are shown without the first pressure cycle, during which some pores become**
**permanently closed. Normal-to-layering flow data are shown including the first pressure cycle. For**
**Bowland shale, flow normal to layering is slower, but for Haynesville shale there is little effect, except that**
**pressure sensitivity is less for flow normal to layering.**
Flow normal to layering in shales is often much slower than flow parallel to layering, but not always. Layer-
normal flow was therefore measured for these rocks using shorter samples than for flow along the layering, and
only at 10 MPa argon pore pressure (Fig. 4). However, for Haynesville shale the direction of flow makes little
difference, except that pressure sensitivity is reduced for layer-normal flow, as would be expected if flow parallel
to the layering is dominated by low aspect ratio, crack-like pores that are relatively compressible. The different
pressure sensitivities of permeability mean that (after the first pressure cycle) flow along the layering becomes
faster at low effective pressures, but slower at higher effective pressures. Bowland shale shows a small reduction
in permeability for flow normal to layering relative to parallel to layering (post the first pressure cycle), and there
is also some indication of a reduced pressure sensitivity, although the dataset is small.





**4.1.4 Storativity of the rocks**
Oscillating pore pressure permeametry yields a dimensionless permeability parameter $\eta$ and a dimensionless
storativity parameter $\xi$ (Eq. (1)), which is the ratio of the accessible pore volume in the rock to the downstream
reservoir volume. A plot of experimentally measured log gain vs signal phase shift angle lies along a line of constant $\xi$
if the sample storativity is constant (Fig. 5). Thus the effective (conductive) porosity of the sample during the course
of the experiment can be calculated. The conductive porosity of many rocks is smaller than the total porosity.
The total porosity also corresponds to a particular value of $\xi$. If all of the porosity were to be involved in the flow,
these $\xi$ values will be equal. Note that a value of $\xi = 1$ corresponds to the downstream volume of the apparatus being
equal to the pore volume of the rock sample. A storativity can also be calculated from data from elastic pore
compressibility measurements. Hasanov et al. (2019) calculated storativity in these two ways.
Figure 5a shows log gain vs phase angle data for Haynesville shale for flows both parallel and normal to layering.
Figure 5b shows corresponding data for Bowland shale and Pennant sandstone, but insufficient data was obtained for
Bowland shale normal to layering, given its much lower permeability. For flow along the layering, both of the shale
types show $\xi < 0.1$, corresponding to the conductive porosity being much smaller ($< 1\%$) than the total porosity of the
rocks (respectively 4.5% and 9.3%). Thus whilst the bulk of the pore space can contribute to gas storage, only a very
small fraction of well-connected porosity contributes to gas flow along the layering in the shales.
The log gain vs phase angle data was non-linear least-squares fitted to obtain an average value for $\xi$ for each rock
type, subject to the constraint that $\xi$ is constant. For Haynesville shale for flow across the layering $\xi$ lies along the
trend $\xi = 0.39$, corresponding to a conductive porosity of ~6.0%. Thus flow across the layering 'sees' more of the
total porosity than flow along the layering, though still substantially less than the amount of total porosity. Whitby
shale (Mckernan et al., 2017) displays the same effect. In marked contrast, for the Pennant sandstone $\xi = 2.72$. This
is close to the value of $\xi = 2.67$ corresponding to the total porosity (4.6%) of the rock, implying a high degree of
connectivity between the pore spaces in Pennant sandstone.

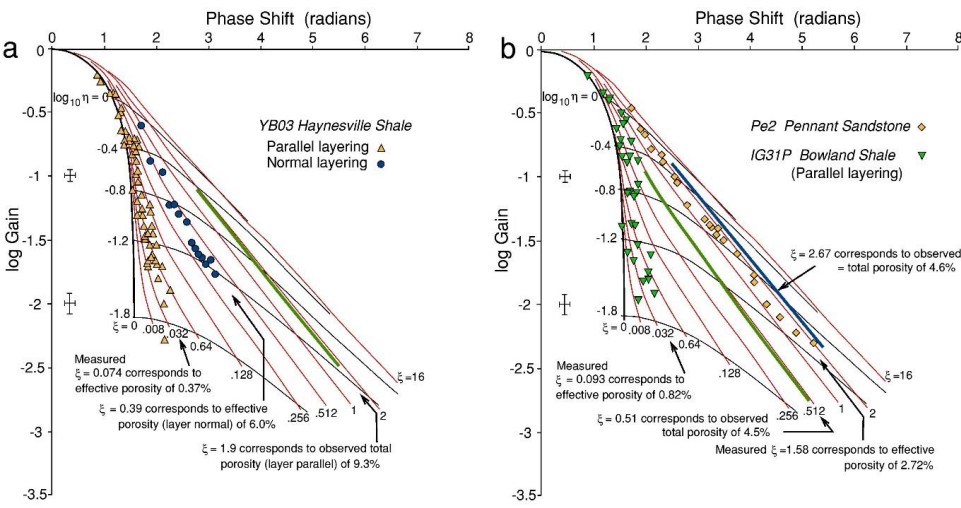





**Figure 5. Log gain vs phase angle data from oscillating pore pressure measurements on :**
**(a) Haynesville shale. $\xi = 1.9$ would correspond to total porosity 9.3% for flow in the sample parallel to**
**layering if all porosity participates in the flow. Observed $\xi = 0.39$ normal to layering is much greater**
**than parallel to layering $\xi = 0.074$, but both are substantially less than that corresponding to total**
**porosity. Flow parallel to layering only 'sees' or 'uses' about 4% of the total pore space, and normal to**
**layering about 42% of the total pore space.**
**(b) Bowland shale. $\xi = 0.51$ would correspond to total porosity 9.3% for flow in the sample parallel to**
**layering if all porosity participates in the flow. Observed $\xi = 0.093$ for flow parallel to the layering**
**corresponds to a conductive porosity (0.82%) much less than total porosity. In contrast, data for**
**Pennant sandstone show observed $\xi = 1.58$ to be closer to that $\xi = 2.67$ which corresponds to the**
**total porosity of the rock.**
**4.1.5 Bulk moduli of compressibility for Pennant sandstone**
Bulk modulus of porosity $K_\phi$ (defined in Eq. (2)) and its effective pressure sensitivity can be measured from the
volume of argon expelled from the rock during increments of confining pressure at constant pore pressure, and
$K_{dry}$ can be calculated using Eq. (4) (Fig. 6a). $K_o$ is the mineral bulk compressibility estimated as the VRH
average at zero porosity (given for these rocks in Table 1).
$K_\phi/\phi$ is the value of the pore bulk modulus referred to the total volume of the rock, rather than to the pore space
volume. $K_\phi/\phi$ and $K_{dry}$ versus Terzaghi effective confining pressure are shown in Fig. 6 for Pennant Sandstone.
$K_{dry}$ is asymptotic to $K_o$ (41.5 GPa) at high pressure.
The pore pressure coefficient $m$, describing the effects of pore pressures on elastic distortions of a porous rock,
and defined in $P_{eff} = P_c(1 - m\,P_p)$ is given in terms of the bulk moduli $K_{dry}$ and $K_o$ in Eq. (7). In Fig. 7 the resultant
$m$ versus effective pressure curves are shown for both Pennant sandstone and Haynesville shale. Bulk moduli are
isotropic properties with values unaffected even when the aggregate displays preferred orientation (shape and
crystallographic) of constituent grains (Andrews, 1978; Mendelson, 1981).

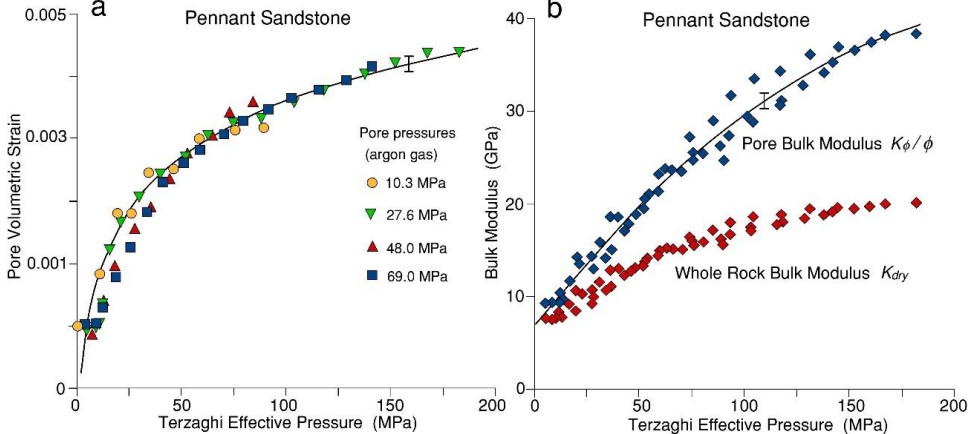


**Figure 6.**



(a) **Volumetric strain (with respect to whole sample volume) for Pennant sandstone at four different constant gas pore pressures. There is no significant effect of magnitude of pore pressure. About 20% of the total pore volume is elastically reduced over a range of 200 MPa effective pressure.**

(b) **Pore bulk modulus $K_\phi /\phi$ from gas expulsion data in (a) for Pennant sandstone, and whole rock bulk modulus calculated from $K_\phi /\phi$ and $K_o$ (41.5 GPa). Pore spaces become rapidly less compliant as effective pressure increases.**

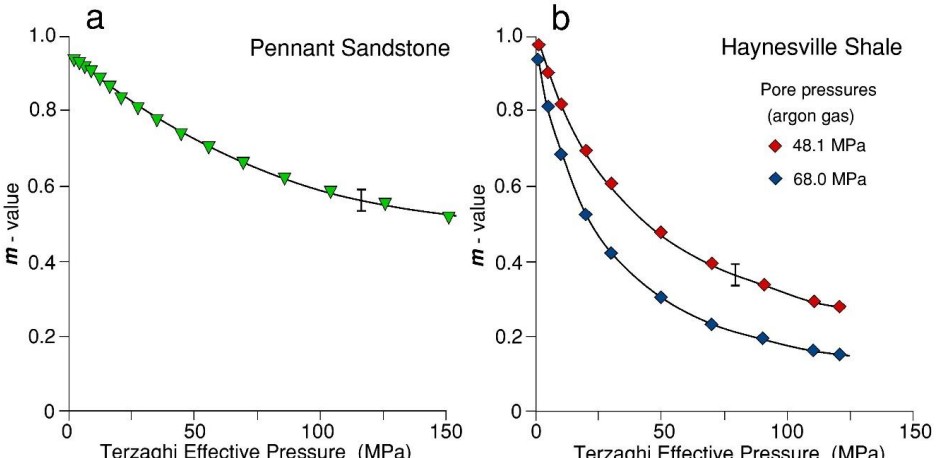

**Figure 7.**

*m* **from bulk modulus data and Eq. (7) for (a) Pennant sandstone and (b) Haynesville shale. The decrease of *m* with $P_{\text{eff}}$ arises from the stiffening of the pore spaces with effective pressure, and the effect is greater for the shale than for the sandstone.**

At low pressure $K_{dry}$ is much less than $K_o$, hence *m* approaches 1. As $K_{dry}$ increases with pressure it approaches $K_o$, hence *m* decreases with pressure, and will eventually reach zero when all pore space has collapsed. Any small increase of $K_o$ with pressure has been ignored (e.g. Calderón et al., 2007). The variation of *m* with pressure forms the basis for describing the decrease in permeability observed as effective pressure increases.

**4.1.6 Bulk moduli of compressibility for Haynesville shale**

Pore volumometry by the expelled gas volume method during progressive increase in confining pressure was carried out on the two shale samples used (Fig 8). The resolution of the pore volume change data is poor because the specimen size was rather small (1.9 cm long). The rapid increase in slope translates to a rapid rise of calculated $K_{\text{dry}}$ compared to Pennant sandstone, until it is a substantial fraction of $K_o$ (61 GPa). However, the total amount of gas expelled corresponds to a closure of about 2% of the initial porosity (0.15% of the whole sample volume). Figure 7b shows pore pressure coefficient *m* calculated from the pore volumometry. *m* decreases rapidly because the $K_{\text{dry}}$ value rises rapidly to become a substantial fraction of $K_o$. It is not clear why the measurements at two different pore pressures are so different, but it is thought to be attributable to different degrees of gas trapping in poorly connected pore spaces.



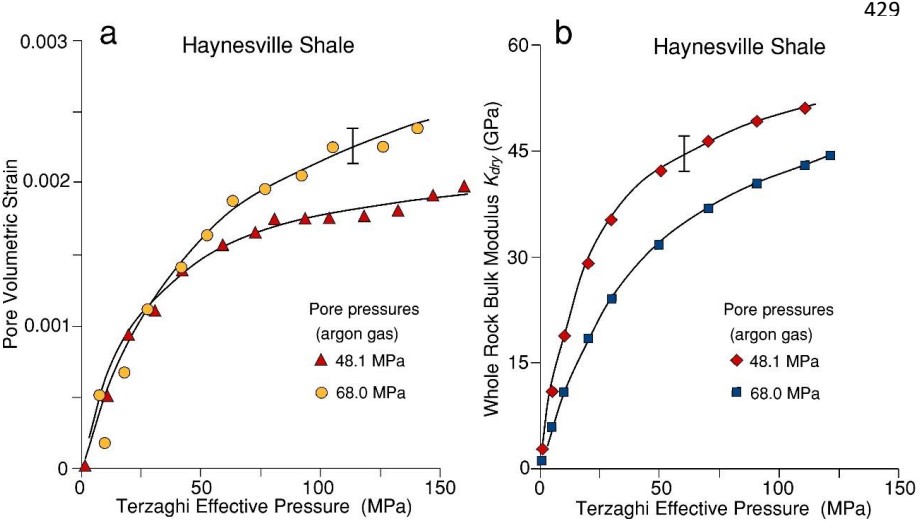

**Figure 8.**

**(a)** **Pore volumetric strain (as fraction of total specimen volume) vs Terzaghi effective pressure for**
**Haynesville shale at the pore pressures indicated. Pore volume loss is approx. only 2% of the**
**initial pore volume of the rock. Logarithmic fits to two of the data sets are shown.**
**(b)** **The gradients of the fitted lines in (a) correspond to the pore compressibility, and were used to**
**obtain $K_{dry}$ vs $P_{eff}$, as shown in (b) for the two pore pressures used. $K_o$= 61 GPa .**

**4.1.7 Bulk moduli of compressibility for Bowland shale.**

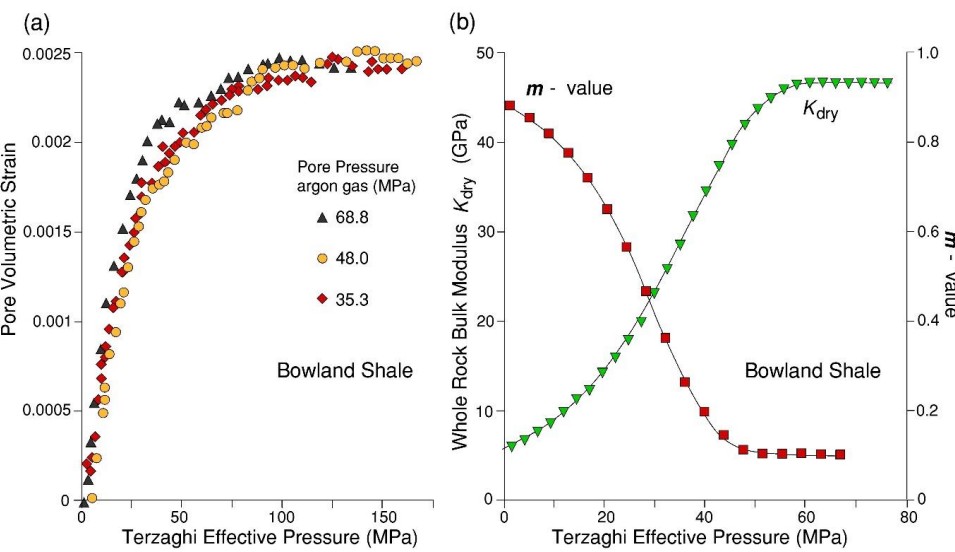

**Figure 9.**
**(a) Pore volumometry of Bowland shale at 34.5, 48 and 69 MPa gas pore pressure. There is no significant**
**difference at the three pore pressures, so that a single polynomial function can be fitted to all the data. The**
**slope of the curves corresponds to the pore compressibility, which decreases markedly with increasing**
**effective pressure.**





**(b) Shows calculated $K_{dry}$ bulk modulus of the sample (pore pressure = 69 MPa) from pore volumometry**
**measurements (inverted triangle symbols). $K_o$ = 52.8 MPa. Also plotted is the m value from pore**
**volumometry (square symbols) for Bowland shale at 69 MPa pore pressure.**
A large specimen (25 mm diameter and 50 mm long) was used for these measurements on Bowland shale, cored
parallel to the layering. Because this is a low permeability rock, a 2 mm thick longitudinal slab of porous sintered
stainless steel was deployed as described earlier, to facilitate gas flow between the rock pores and the pore
pressure system. During pressure cycling it was necessary to correct data for the storativity of this plate. Figure 9
shows pore volumometry at 34.5, 48.1 and 69 MPa MPa argon gas pore pressure and $K_{dry}$ data for Bowland shale.
Measurements were very reproducible and, unlike the Haynesville shale sample, there was no significant effect of
the magnitude of the pore pressure used. The amount of gas expelled during an effective confining pressure cycle
of 150 MPa corresponds to closure of ~8.4% of the initial (4.5% porosity) pore space, or about 0.04% of the total
rock volume. As also observed for Haynesville shale, this represents a very small fraction of the total porosity.
The poroelastic coefficient $m$ calculated from the volumometry data is shown in Fig. 9b. Like the Haynesville
shale, the poroelastic coefficient obtained from pore volumometry decreases substantially with Terzaghi effective
pressure but does so at a similar rate to the Haynesville shale.

**5.   Discussion**
**5.1 Generation of pore pressure during undrained loading**
If drainage channelways become constricted during application of increments of $P_c$ whilst $P_p$ is also high, the
rock might become effectively undrained and hence pore pressure increments can arise. The magnitude of an
induced pore pressure under undrained conditions can be estimated from the Skempton parameter $B$, where
$$dP_p(\text{induced}) = B dP_c = \frac{C_{PP} + C_0}{C_{PP} + C_f} dP_c \qquad (10)$$
$B$ is the Skempton $B$ parameter of soil mechanics (Lockner and Stanchits, 2002). $C_{pp}$ is the compressibility of the
pore space arising from a change in pore pressure, and is usually much less than the compressibility of the pore
fluid $C_f$. Thus $B$ will lie between 0 and 1.0. Because usually $C_{pp} \gg C_o$ (where $C_o = 1/K_o$ ),
$$B \approx \frac{C_{PP}}{C_{PP} + C_f} = \frac{1}{1 + \frac{C_f}{C_{PP}}} \qquad (11)$$
For a gas saturated rock $C_f > C_{pp}$, hence $B \to 0$, and a gas-saturated rock will therefore never develop appreciable
pore pressures, especially at high porosities and from low initial gas pressures even when undrained, hence was
not considered to be an issue in the present experiments.
For a liquid-saturated rock however, this will not be true. $B$ will approach 1 when $C_{pp} \gg C_f$. For liquid-
saturated porous sandstones under hydrostatic loading, Green and Wang (1986) found that under undrained
conditions, induced pore pressures were close to the applied confining pressures over a range of 60 MPa confining
pressure, thus the mean externally applied stress is almost totally transferred to the pore fluid via the
compressibility of the pore spaces.



The time constant for the dissipation of excess pore pressure in a region of characteristic dimension $L$ in a
material of permeability $k$ is on the order of

$$t = \frac{\phi \mu (C_f + C_{PP}) L^2}{k} \qquad (12)$$

$t$ is the time required for pressure to decay by factor 1/e at distance $L$. The ratio $k / \mu (C_f + C_{pp})$ is the hydraulic
diffusivity $\kappa$ (dimensions m²/s). For water, viscosity $\mu$ is 0.001 Pa s. Taking the bulk modulus $K_f$ (= 1/fluid
compressibility, $C_f$) to be 2 GPa, and the permeability to be $10^{-18.5}$ m² for Haynesville shale at about 5 MPa
effective pressure (this is the highest permeability measured, which would apply after an excess fluid pressure had
been generated by compaction), $\kappa \sim 2 \times 10^{-6.5}$ m²/s. This leads to $t = 30$ sec for $L = 2$ cm. Time $t$ is shorter by a
factor 1/30 when the pore fluid is gas owing to its lower viscosity (Gosman et al., 1969). This equation is for
constant $k$, but when $k$ is a strong function of $P_{eff}$, decreasing perhaps 300-fold at high effective pressures, up to 5
minutes may be required for small pore pressure transients to decay.
**5.2 Simple model for pressure-dependence of permeability**
The simplest approach to describing the influence of pore space geometry and connectivity on permeability is to
regard the pores as a bundle of circular capillary tubes, so that the equation for viscous Poiseuille flow can be
applied and permeability calculated as a function of capillary tube radius. The circular capillary tube is a special
case of flow through tubes of elliptical cross section. In this case the flow rate then becomes acutely sensitive to
the short radial dimension of the tube, and the more eccentric the tube cross-section the greater will be the
sensitivity of its shape to externally applied effective pressure (Seeburger and Nur, 1984). Ma et al. (2018) imaged
connected pores spaces in shales, including Haynesville shale, as thin, crack-like shapes lying parallel to bedding
and of nanometric widths. Such pores in shales are not identical to straight capillary tubes of elliptical cross
section, but we can explore the extent to which the pressure sensitivity of observed permeability can be modelled
as such (Mckernan et al., 2017).
For a single tube of long axis $2c$ and short axis $2b$ the volume flow rate $q$ of a fluid of viscosity $\mu$ along a
hydraulic pressure gradient $dP_p/dx$ is well known to be

$$q = \frac{\pi}{4\mu} \left( \frac{b^3 c^3}{b^2 + c^2} \right) \left( \frac{dP_p}{dx} \right) \qquad (13)$$

and for $N$ parallel tubes embedded in an elastic matrix of volume $V$ and intersecting a 1 m² area normal to their
length the total flux $Q = Nq$. Separating out the viscosity and pressure gradient, the permeability $k_o$ of the array is
$k_o = (N \pi /4) (b^3 c^3/(b^2 + c^2))$.   Dimension $c$ does not change with externally applied pressure for the elliptical
crack, whereas for the tapered crack it does, such as to keep the aspect ratio approximately constant (Mavko and
Nur, 1978), and Seeburger and Nur (1984) found that there is little difference in the effect of hydrostatic pressure
on flow rate when the tube cross section is elliptical or tapered. In terms of aspect ratio of an assumed elliptical
cross section $\alpha = b/c$, thus

$$k = \frac{N\pi}{4} c^4 \left( \frac{\alpha^3}{1 - \alpha} \right) \qquad (14)$$





The porosity $\phi = N\pi\, b\, c$. Parameters $\alpha$, $c$ and $N$ that satisfy Eq. (4) are non-unique. $N$ can be increased whilst pore
aperture is decreased, keeping $k$ unchanged. A further constraint is therefore required, and this is provided by the
porosity $\phi$, which is already known as a property of the material. Porosity is given by $\phi = N\, c^2\, \alpha\, \pi$. Thus Eq. (14)
becomes
$$k = \frac{\phi c^2}{4}\left(\frac{\alpha^2}{1-\alpha^2}\right) \qquad (15)$$

Applying a hydrostatic pressure $P$ to a solid bearing elliptical cracks reduces the $b$ dimensions of all pore spaces,
and hence reduces the hydraulic transmissivity. The spatial density of the ellipses is assumed to be sufficiently
small that the elastic strain fields of each do not interact significantly. From Seeburger and Nur (1984), following
Walsh (1965) and Mavko and Nur (1978) the bulk modulus $K_{dry}$ of a solid of volume $V$ containing $N$ tubular
cracks of elliptical cross section and semi-major axis $c$ is given by
$$\frac{1}{K_{dry}} = \frac{1}{K_0} + \frac{1}{K_0}\left[2Nc^2 d\,\frac{1-v^2}{1-2v}\right]$$

Thus
$$\frac{K_0}{K_{dry}} - 1 = 2Nc^2 d\,\frac{1-v^2}{1-2v} \qquad (16)$$

$d$ is the elliptical section tube length in the third dimension ($= V^{(1/3)}$).
Taking  $m = (1 - K_{dry}/K_o)$, the left hand side is  $m/(1-m)$, and the expression can be rearranged with $c^2$ on the
left side:
$$c^2 = \left(\frac{m}{1-m}\right)\left(\frac{1-2v}{1-v^2}\right)\frac{1}{2Nd} \qquad (17)$$

This can replace $c^2$ in Eq. (17), to give :
$$k = \left(\frac{\phi}{8Nd}\right)\left(\frac{\alpha^2}{1+\alpha^2}\right)\left(\frac{m}{1-m}\right)\left(\frac{1-2v}{1-v^2}\right) \qquad (18)$$

$m$ is measured by pore volumometry as a function of Terzaghi effective pressure hence $k$ is a function of effective
pressure. For $b<<c$ it is primarily the reduction of the $b$ dimension with increasing pressure that reduces
permeability. However, Mavko and Nur (1978) and Seeburger and Nur (1984) showed that the bulk modulus of a
porous solid of given porosity is not affected by the shape (eccentricity) of the pores. All pores change volume by
the same fractional amount. Only the distortion under pressure of the more eccentric ones is likely to affect the
permeability, although all pores will affect the storativity, according to how well connected they are. The
'connected' porosity estimated from the log gain versus phase shift plot, that is much smaller than the total
porosity, is used in Eq. (18). Its small value implies that most of the porosity is not being inflated during the
passage of the pore pressure wave, hence during the time-scale of the pressure oscillation the greater part of the
porosity is closed off by the action of the effective pressure.
Eq. (18) can be fitted to the permeability data log $k = f(P_{eff})$ measured for rock types studied using the non-linear
least-squares fitting routine Solver in MS Excel, to estimate the parameters $N$, $v$ and $\alpha$. Via the inferred effective





porosity the conductive pore width can also be estimated. The results of the fitting exercise provide the parameters
for a bundle of capillary tubes that *behaves in the same way* as the measured rocks.  This is not to say that the
geometric arrangement of a simple capillary tube bundle corresponds to the pore space configurations in these
rocks, nor that a solution can be found for all rocks. The pressure sensitivity lies in the function that describes *m* as
a function of pressure, obtained from pore volumometry, and incorporating the effective pressure coefficient *n*.
Figure 10 shows the fit to the data for the Pennant sandstone; fit parameters are in Table 3.

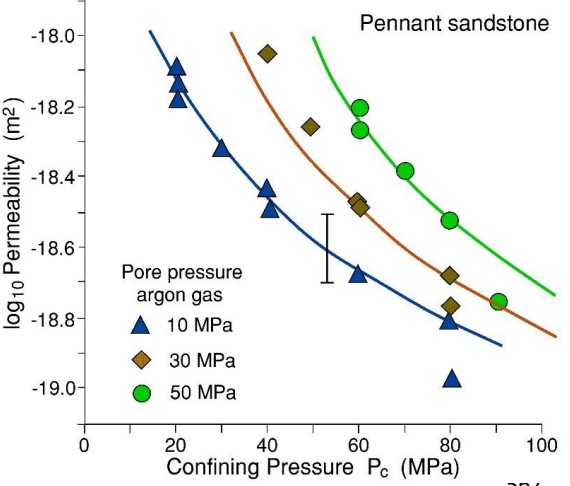

**Figure 10. Gas permeability data for Pennant sandstone normal to layering, for three constant pore pressures. The three continuous curves are for Eq. (18) non-linear least-squares fits to the data.**

Unlike the relatively homogeneous distribution of pore channels in the shales

down to the micron scale, in the Pennant sandstone the greater part of the rock volume is not porous, as it
comprises large quartz and feldspar grains. The 4.6% porosity is contained mostly in the spaces originally between
these grains that are now largely filled with phyllosilicate and oxide phases, i.e. about 26% of the total rock
volume, and is microstructurally in some ways comparable to a shale. Therefore in Table 3 the estimated
conductive channel dimensions are based on flow through this reduced volume fraction.
Figure 11 shows the fits to the permeability data for Haynesville shale. The cross-section shape of the elliptical
tubes is extremely eccentric and the shorter width of the tubes is measured in nanometres. This is consistent with
the observations of the dimensions of connected bedding-parallel porosity in the high-resolution tomography (CT)
observations of Ma et al. (2018) for Haynesville shale.












**Figure 11. Permeability of Haynesville Shale versus total confining pressure for various values of constant gas pore pressure. The curves shown are the permeabilities calculated using the elliptical section pore channels model (Eq. (18)).**


**Figure 12. Permeability of Bowland Shale versus total confining pressure for various values of constant gas pore pressure. The curve and data shown for $Pp = 0.1$ is the effective pressure fit to all the data as shown in Fig. 3, collapsed onto a single least squares best-fit curve ($\log_{10}k = -0.503 \log_{10}P_{eff} -17.26$) for a pore pressure coefficient made to vary linearly with Terzaghi effective pressure according to $n = (1+P_{eff}(MPa)/85)$. Measured data for the separate pore pressures are shown, with best-fit curves with the variable pore pressure coefficient. $n$-values are shown to indicate how they increase from left to right.**



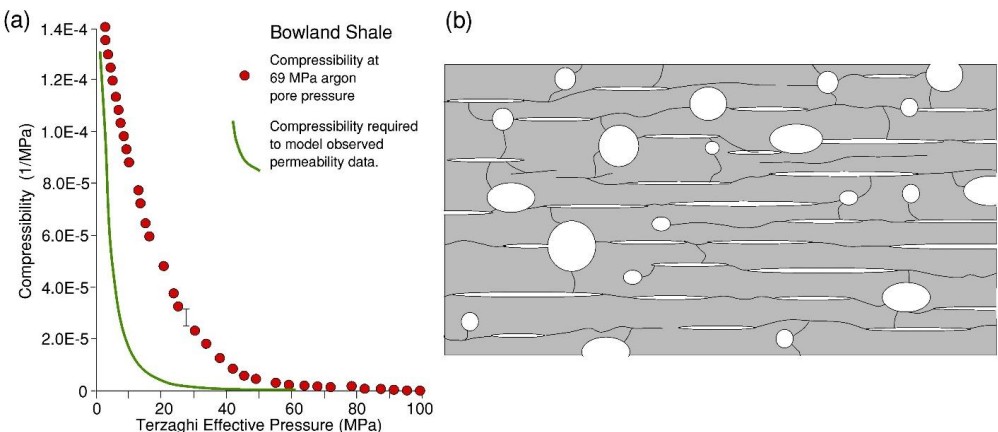


**Figure 13.**
**(a) The experimentally observed variation of pore compressibility at 69 MPa pore pressure (filled circles) vs**
**Terzaghi effective pressure for Bowland shale, derived from the data in Fig. 9. The reciprocal of this**
**compressibility is $\phi/K_\phi$. This rate of reduction of compressibility with effective pressure cannot predict the**
**observed pressure sensitivity of permeability that is observed experimentally. The continuous curve shows**
**what the trend would have to be like in order that the single capillary tube model can behave in the same**
**way as the rock.**
**(b)  Schematic illustration of the porosity model best able to explain the permeability and bulk modulus**
**data in the shales. Highly eccentric pores and cracks lie parallel to layering but are well-connected,**
**accounting for easy gas transport yet using only a small fraction of the porosity. These narrow pores are**
**easily constricted by hydrostatic pressure. Most of the storage capacity resides in the larger, equant pores**
**of dimension about 1 micron that are poorly linked and not easily closed down by hydrostatic pressure.**









**Table 3. Fit parameters for the capillary tubes bundle model applied to describe the permeability of**
**Haynesville shale and Pennant sandstone at low effective pressures, when the permeabilities are not**
**strikingly different. $n$ is the pore pressure multiplier for the permeability data, $N$ is the number of pores**
**intersecting a 1 m² area normal to the flow path, $a$ is the pore shape aspect ratio and $\nu$ is the Poisson ratio.**
**$2b$ is the mean short dimension (nm) of the elliptical cross section and $s$ is the average pore spacing**
**(microns).**

643 -------------------------------------------------------------------------------------------------------------------





| 644 | | Haynesville Shale | Pennant sst |
|---|---|---|---|
| 645 | $n$ | 0.99 | 0.86 |
| 646 | $N$ | 1.03 E+11 m$^{-2}$ | 8.4 E+11 m$^{-2}$ |
| 647 | $\alpha$ | 0.0051 | 0.004 |
| 648 | $v$ | 0.17 | 0.10 |
| 649 | $2b$ | 13.5 nm | 21 nm |
| 650 | $s$ | 3.1 μm | 1.5 μm |
| 651 | Conductive porosity | 0.3% | 3.8% |

652 -----------------------------------------------------------------------------------------------------------------


The form of the curve of $K_{dry}$ vs Terzaghi effective pressure does not permit the simple elliptical section
capillary tubes model to be fitted to Bowland shale, because the observed rate of decrease of $m$ with effective
pressure is insufficiently rapid to explain the three orders of magnitude decrease of permeability observed over
this pressure range (see Fig. 12). Figure 13a compares the observed variation with effective pressure of pore
compressibility factor $m$ to the variation that would be required to be able to make such a fit. It is inferred that
pressure must be able to act in this rock to close down pore connectivity in one or more additional ways to the
elastic compression of elliptical channel cross-sections. These could involve development of increased tortuosity
of channelways, or the existence of a more complex distribution of connected pores of different sizes and shapes.
The simple model of a set of similarly-sized and shaped channels that can behave in a comparable way to a real
pore network is clearly inapplicable to this rock.
**5.3 The effective pressure coefficients, *m* and *n***
In the context of permeability, $n$ is the multiplier of pore pressure in the definition of the modification of
Terzaghi effective pressure that brings observed permeability data at different constant pore pressures onto a
common curve (e.g. Fig. 12), thus $P_{eff} = P_c - nP_p$. $n$ takes a value close to unity in the case of the experimental
data on Haynesville shale and Whitby shale, and 0.86 in the case of Pennant sandstone. In other studies, observed
departures from unity have been attributed to, for example, differences in the roles of elastically stiff and
elastically soft mineral components surrounding the pore spaces in responses to changes in $P_c$ relative to changes
in $P_p$ (e.g. Zoback and Byerlee, 1975; Kwon et al., 2001; Ma and Zoback, 2017), resulting in different rates of
change of pore volume with $P_c$ and $P_p$.
On the other hand, in Eq. (4), for a homogeneous, isotropic elastic matrix, it is the value of $K_{dry}$, the bulk
modulus of the porous rock, that determines the change in geometry of pore spaces, and hence permeability, in
response to effective pressure change. The theoretical expression for the effective pressure coefficient $m$ for elastic
*deformations* of a mechanically linear, homogeneous and isotropic rock is given by Eq. (7) and this parameter
appears in the expression for the permeability according to the bundle of capillary tubes model (Eq. (18)). Using
the pore fluid displacement method (Figs. 6, 8 and 9) we have found that in all cases $m$ decreases from near unity
with Terzaghi effective pressure according to the pressure dependence of $K_{dry}$, whereas for Pennant sandstone and





Haynesville shale, observed $n$ remains close to unity for permeability data and exceeds unity for Bowland shale
over Terzaghi effective pressures from zero to *ca* 80 MPa, thus $m \neq n$. Nur and Byerlee (1971) took care to point
out that $m$ as defined in Eq. (7) cannot generally be used as a predictor of effective pressure coefficient for
particular processes, like permeability, mechanical strength and elastic wave velocities, even though all involve
elastic distortions.
As was pointed out earlier, pressure sensitivity of permeability according to the simple capillary bundle model
cannot behave in the same way as was observed experimentally for Bowland shale. Also, a single value of $n$
cannot reconcile permeabilities at different pore pressures for this rock. Figure 12 shows the permeability data for
Bowland shale separated into measurements at different pore pressures. By extending the collective fit between
log permeability and effective pressure shown in Fig. 12 to the data at each pore pressure, the downward
divergence of the curves becomes apparent.  This can be described empirically by fitting a linear variation of $n$
with Terzaghi effective pressure, such that $n = 1$ at low effective pressures, rising to $n = 1.6$ at the upper end of the
pressure range used. This is interpreted as a further manifestation of the pore structure complexities that mean that
this Bowland shale cannot be described by a simple capillary tube bundle model.
**5.4 Relationship between observed pressure-dependent permeability and mineralogy**
Several studies have reported the relationships between mineralogy of shales and related rocks and their
petrophysical properties (e.g. Kwon et al., 2004; Ma and Zoback, 2017). The rocks used in this study display a
spectrum of mineralogy that is reflected in their permeabilities, both in terms of absolute values and their
sensitivity to effective pressure.
Pennant sandstone is typical of tight gas sands in which the load bearing framework is of continuous quartz and
feldspar grains with what would otherwise be a large porosity that is mostly filled with some detrital muscovite
plus diagenetically-introduced clay and oxide phases (Wilson and Pittman, 1977; Howard, 1992). Prior to the pore
filling there was a degree of intergranular pressure solution and formation of quartz overgrowths around quartz
grains.  The protective armour around the filled pore spaces afforded by the quartz framework is thought to have
limited degree of compaction of the pore filling, in which most of the present porosity resides. Relative to the
volume of the inter-quartz spaces, the porosity of the filling would be ~20%, and it is thought that this contributes
to the relatively high overall permeability and reduced pressure sensitivity of Pennant sandstone.
The Bowland and Haynesville shales are mineralogically and microstructurally strikingly different. It is
important to remember that these are particular samples taken from their respective sequences and may not be
especially representative of their host sequences at all. The Bowland shale sample is a phyllosilicate-rich,
carbonate-poor siliceous mudstone with sufficient phyllosilicate to form a contiguous matrix, and this is likely to
be responsible for the relatively low bulk modulus (53 GPa) of the rock and hence low permeability. The
Haynesville shale is a carbonate-rich (>50vol%), phyllosilicate-poor siliceous mudstone with a higher bulk
modulus (61 GPa). The carbonate grains (fossil fragments and diagenetic carbonate) provide a stiff framework of
contiguous grains, helping to maintain open porosity and to resist its elastic compaction. Despite these qualitative
observations that can be made about how mineralogy and microstructure impacts upon permeability, the present
results do not form a basis for making any quantitative correlations.



### 5.5 Inference of key characteristics of pore space geometry in shales

Much has been written on pore space geometry based on SEM, TEM and Xray CT imaging of shales, but important characteristics can be inferred from observations of bulk petrophysical properties. Key points noted in the present study are:

● The storativities for both shales are extremely small for flow paths lying parallel to the layering, such that over 90% of the available pore space is not participating in the flow.

● At low effective pressures, the permeabilities of all three rocks are similar, but with increasing effective pressures they diverge at markedly different rates. Marked sensitivity of permeability to effective confining pressure implies that conductive (well-connected) pores are flat and crack-like. This is supported by permeability modelling, that suggests that for a bundle of elliptical-section capillary tubes of equivalent permeability behaviour, their aspect ratios are extremely small and the narrow dimension is expected to be in the nanometric range (Table 3).

● For flow normal to layering, at least in Haynesville shale, storativity is much greater than for flow across the layering, but still implies that over half of the pore space is not participating in the flow.

● Permeability in both shales is very low under elevated effective pressures compared to Pennant sandstone, which is of similar overall porosity, implying that connected pore spaces are narrow and/or poorly connected/tortuous.

The above observations suggest that the effective configuration of pores spaces corresponds to the sketch shown in Fig. 13b, with a population of highly oriented, crack-like pores parallel to layering that account for only a small fraction of the total porosity but dominate the hydraulic transmissivity through the rock mass parallel to the layering and also account for the low storativity associated with flow along the layering. These are poorly connected to larger, probably more equant pores by conduction channels trending across the layering, and which contain most of the gas storage space in the rock. The equant pores are 'seen' more easily for flow across the layering, so that this flow is characterised by higher storativity, as demonstrated for Haynesville shale. Such storage pores are likely to be much slower to drain (or to fill) in response to an applied pore pressure gradient than implicit in the laboratory-measured permeability data. This suggests that permeabilities measured by transient flow methods in the laboratory may lead to an over-conservative estimate of the potential for drainage of a gas reservoir in shale, and perhaps help partially to explain the long-term persistence of flows from some shale gas reservoirs (e.g. Guo et al., 2017; Wang, 2017).

### 6. Conclusions

Permeabilities as functions of effective pressure were measured using the oscillating pore pressure method at 20 ºC for three rocks (Haynesville and Bowland shales and Pennant sandstone) of low permeabilities and comparable porosities. Tests were at effective pressures ranging up to 90 MPa with argon gas as permeant. From exhibiting comparable permeabilities at low pressures they diverged markedly with increasing pressure. Pennant sandstone showed permeability reduction with pressure of less than ten-fold, Haynesville shale became less permeable by almost two orders of magnitude, whereas Bowland shale was reduced in permeability by more than 3 orders of magnitude. The different pressure sensitivities of permeability correlated inversely with their (pressure sensitive)





bulk moduli and qualitatively with mineralogical differences, going from a continuous framework of stiff quartz
grains (sandstone) through a carbonate-rich framework (Haynesville shale) to a contiguous matrix of phyllosilicate
grains (Bowland shale).
High storativity of the sandstone implied that most of the available pore space was involved in the gas flow, but
in the shales, for flow parallel to the layering, less than 10% of the available pore space was involved in the flow.
For flow in the Haynesville shale across the layering a larger pore space fraction was involved, but still much less
than all the available pore space. Thus only a small fraction of the total pore space can be inferred to be well
connected in the shales. This implies that whilst the permeability we measure in the oscillating pore pressure
experiment is that associated with gas transport through the rock mass, a lower effective permeability applies to
the ability of the gas to flow into and out of the storage pores.
A simple model of permeability was developed based upon connected pore space behaving in a way similar to a
bundle of capillary tubes of highly eccentric cross section. By fitting the model to the experimental data, it was
possible to demonstrate that this model behaved in a similar way to the rocks for the case of Pennant sandstone
and Haynesville shale, but the model could not behave in a way compatible with the marked pressure sensitivity of
permeability for the Bowland shale. It was inferred that a more complex distribution of connected pore spaces of
varying dimension and tortuosity would be required to behave like the Bowland shale sample.
**Author contribution**
EHR was responsible for the conceptualization and methodology of the study, carrying out the bulk of the
experiments, compilation and analysis of data and writing the manuscript. JM was responsible for the acquisition
and management of financial support, carrying out the FEM analysis, contribution to experimental design and data
presentation, and preparation of the paper. YB carried out the experiments on Haynesville shale under the
supervision of JM and EHR as part of his doctoral research.
**Data availability**
All of the experimental data acquired in this research is freely accessible and collated in supporting datafile
DF1.csv. In correspondence with UK Research Council requirements is deposited in the UK National Geoscience
Data Centre, identified by the title of this paper. It is also downloadable from https://zenodo.org/record/5675601
The Authors declare that they have no conflict of interests.

**Acknowledgements**
This work was supported by UK Natural Environment Research Council grant NE/R017883/1 and was part of
the Challenge 2 NERC Unconventional Hydrocarbons program. Y. B. was supported for a postgraduate research
studentship by the Petroleum Technology Development Fund - Nigeria.
Sections of borehole core of Bowland shale were kindly provided by Igas, and of Haynesville shale by BG
International, now Shell. X-ray diffraction characterization of test materials was carried out by John Waters
(University of Manchester). Total Organic Carbon measurements of Haynesville shale were carried out by Geir
Hansen of Applied Petroleum Technology AS (Norway). GKN sinter metal filters GmbH kindly donated the 2mm



thick SIKA R1AX porous stainless-steel plates used in this work. Experimental Officers Stephen May and Lee
Paul contributed to equipment maintenance. Mike Chandler and Rochelle Taylor provided helpful discussions.

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
