# Peer review of "Matrix gas flow through 'impermeable' rocks - shales and tight"

_Solid Earth, 2021_

## Author Response (AR2)

Responses to referees' and correspondent's comments:

Anonymous referee RC1 : posted 4 January 2022

**General Comments**

The paper presents interesting study of permeability of tight rocks with comparable porosities over a wide range of pressure conditions. The authors explain the pressure dependence of permeability using a model that capillary tubes with eccentric cross sections. However, the model fails to explain the permeability evolution in Bowland shale which is interpreted to be a result of the heterogeneity of the pore size and tortuosity. Overall, I found the study well supported by the amount of data and nicely discussed.

**Specific Comments**

1. In line 88-89, it is not clear what the author means by 'bedding horizontal'. Please clarify.
2. The author mentioned conducting permeability measurements using both pulse-decay and oscillation method. But it is hard in the later plots to distinguish measurements from different method. Could the author elaborate on how much difference would it make using different method for permeability measurements in this study.
3. In line 221-225, as described by the author, the samples were exposed to higher effective pressure before the application of pore pressure. Would this contribute partly to the later observed difference in the 'm' and 'n' variation in the effective pressure law.
4. In line 344-349, would the bioturbation in the Haynesville shale be partly the reason for the higher permeability and lower pressure sensitivity in the normal to bedding flow?
5. The authors attributed the different pressure sensitivity of permeability in the Bowland shale to the pore structure complexities (heterogeneity of the pore size and tortuosity). Is there any direct microstructural evidence comparing the Bowland shale to the other two rock types?
6. The use of pore pressure parameter 'n' and the pore pressure coefficient 'm' in the discussion of the effective pressure can be confusing. It might help the reader if this is introduced and discussed earlier in the paper (line 258-260 might be a good place for a clarification).

**Technical Corrections**

1. In section 2, it might be more reader friendly and easier to compare if the author could put the composition proportions, density and porosity data in a table.

**Authors' responses:**

The referee is thanked for complimentary general comments and specific comments of a constructive nature. The latter are addressed as follows, in order of the six numbered points:

1. The bedding trace in the images of Pennant sandstone microstructure are 'horizontal' with respect to the page orientation. To avoid confusion, in the text we now say that the bedding trace is parallel to the long axis of the images.

2. We have previously reported (McKernan et al. 2017) excellent agreement between the results of pulse transient decay and oscillating pore pressure methods for permeability measurements on the same material at the same conditions, and we have now made it clear in the text.
3. Concerning the sequence of application of confining and pore pressures, in a sequence of measurements the aim is initially to apply an effective pressure equal to the maximum the rock will experience in that sequence of tests, to 'pre-condition' the rock, otherwise the sample will not display recoverable elastic behaviour in successive pressure cycles. Obtaining reproducible response to repeated pressure cycling was essential to obtain meaningful permeability/pressure relations. This is not thought to contribute to differences in the pore pressure coefficients $m$ and $n$, which describe physically different behavioural responses.
4. The different permeabilities and pressure sensitivities of Haynesville shale according to orientation must be reflections of the compressibilities and pressure sensitivities of connected pore spaces, and ultimately this must be relatable to microstructural anisotropy. It seems quite likely that differences like these could relate to factors like bioturbation, but unfortunately we do not have any data on this beyond reasonable speculation.
5. There is a substantial amount of microstructural data available on the Haynesville shale, on samples taken from the same cores and close to those used in this study. The study of Ma et al. (2018, op.cit.), using a range of different imaging techniques over a range of scales, shows that the conductive pore dimensions are of the same order of size as calculated from the simple modelling used in this paper. Dowey and Taylor (2020, op.cit.) provide details of the range of diagenetic features represented in this shale. Ma et al. (2019, Energy, 18, 1285-1297) provide further detail on microstructures. The Bowland shale is mineralogically and microstructurally highly heterogeneous over short distances, and there have been detailed studies of such variability  reported (Fauchille et al. 2017, Marine and Petroleum Geology, 86, 1374-1390, and Fauchille et al. 2018, Marine and Petroleum Geology, 92, 109-127) but unfortunately not from the particular phyllosilicate-rich, carbonate-poor core-section studied here. Thus we cannot say more about the source of the differences between these rocks other than attributing differences to generalized mineralogical and microstructural contrasts. This issue is commented upon in the discussion (sections 5.4 and 5.5).
6. We have included a clarification of the different significance of the two pore pressure coefficients at the line suggested and also at line 316.

Technical point 1:  Table 1 already contains for each rock type the mineralogic composition, and elastic moduli data. Because other data for different rock types is reported in slightly different ways, we considered including it in the paragraphs describing the characteristics of each rock type might be clearer.
* * *
Comments from Prof Christian David, posted 20 Dec. 2021

**General comments**

This is a very interesting paper focusing on the effect of effective pressure on the permeability and the deformation of tight rocks, two shales and one tight sandstone. The

pressure sensitivities of the selected rocks were interpreted through poroelasticity theory, and the link with the expected evolution of microstructures in these tight rocks is tentatively given. The authors propose a simple model made of a bundle of capillary tubes with elliptical cross-sections to account for their observations on the permeability decrease with pressure. Another interesting outcome of this work is the discussion on the effective pressure coefficients, with a comparison between the "m" value used in poroelasticity (the so-called Biot coefficient) and the "n" value derived from permeability vs. pressure evolution. It turns out that one of the rocks, the Bowland shale, behaves quite differently compared to the other rocks, and this is explained by strong contrasts in the microstructures and mineralogical content.

**Specific comments**

When reading the paper, it is clear that the data obtained by the authors are of very high quality. This allowed them to analyze thoroughly their data set on the basis of existing theories or models. The outcome is quite convincing and provides a strong basis for future studies on the transport properties in tight rocks. Nevertheless there are some points which could be clarified to my viewpoint:

1. Concerning the velocity anisotropy defined at line 100, I don't understand how this relates to what the authors call "15.5% axial and 3.1% radial" for the Pennant sandstone.
2. When discussing the pressure sensitivity of $K_o$ at line 267, why only providing the law for quartz? Is the conclusion (i.e. negligible pressure sensitivity) the same for all the other minerals including phyllosilicates?
3. In the section 4.1.1 at line 280, the authors check several permeability vs. pressure laws like k vs. $P_c$ (linear) or log k vs. $P_c$ (exponential), but how about log k vs. log $P_c$ (power-law)? The exponential law (i.e. linear fit of log k vs $P_c$) is relevant for other tight rocks like for the Grimsel granodiorite (benchmark KG²B in which the first author had participated), so the observed non linearity seems to be typical of the shaly rocks tested here.
4. There is some confusion I think in the definition of Terzaghi effective pressure. In line 241 the Terzaghi effective pressure is defined as $P_c - P_p$ as it should be, but later several times (e.g. line 306) the Terzaghi effective pressure becomes $P_c - nP_p$. This should be clarified.
5. On the same topic, could the authors explain how they estimate the effective pressure coefficient "n" from the permeability data set? Maybe a new figure may help to explain.
6. At line 398 the definition of $P_{eff}$ is wrong.
7. At line 489, in the definition of hydraulic diffusivity the porosity is missing in the denominator. In the same paragraph, the authors discuss the values of the time constant from the diffusivity, assuming that the pore fluid is either gas or water. However, doing so they assume that the gas permeability that they have measured is the same as the water permeability: can we be sure of that? Again in the KG²B benchmark, a significant difference between permeability values obtained with either gas or liquid as the pore fluid were found for the Grimsel Granodiorite.
8. In equations 14 and 15, the denominator should be $1+\alpha^2$. Equation 16 and the one above should be checked as the units don't match. The product $Nc^2d$ should be dimensionless, but it is not the case because N is defined as a number of tubes per unit

surface. As this equation is used to get equation 18, that one is also problematic because 1/Nd should have the unit of m².

Technical corrections

1. At line 63, I wonder if all the digits are significant for the vol%
2. In Table 1, for Bowland is it correct to read for kaolinite 0.26% +/- 2.6%?
3. In Table 1 the density of quartz for Pennant is wrong
4. At line 181, the definition of storativity should be more general, because for the downstream storativity it is not the pore volume that comes in.
5. At line 536 it should be "in Eq. (15)"
* * *
*Authors' responses:*

Prof David is thanked for his helpful and constructive evaluation and comments.

**Specific comments** (8 numbered points):

1. The numbers 15.5% axial and 3.1% radial for seismic anisotropy at room pressure pertain respectively to the ratio of 'axial' (velocity parallel to the axis of a core cut normal to bedding) to mean of velocities measured parallel to bedding, and the 'radial' means the ratio of max to min velocities measured along the core radial directions (parallel to bedding). We agree this statement is unclear, and it has been clarified in the text.
2. Concerning the pressure sensitivity of $K_o$, quartz was taken as an example of a framework silicate and to illustrate its negligible effect over the pressure range of interest. The same is true of other silicate minerals and also phyllosilicates (e.g. Anderson, 2007; Zanazzi and Pavesi, 2002). A further comment with citations has been added at this point.
3. Concerning the form of the relationships used to describe the dependence of permeability on effective pressure, the common use of an exponential law (or other simple functions) can be convenient but purely empirical. They can be useful for interpolation within the range of data described, but may lead to physically unrealistic predictions if extrapolated beyond that range. In this paper we have made use of exponential, power law and polynomial fits to data as convenient tools for the description of the data within their ranges. We have attempted to erect a simple model to describe permeability versus effective pressure based around a simple analogue for pressure-sensitive pore space geometry. In this respect it is a physically-based description of a pore space geometry that behaves, for two of these rocks, in the same or in an analogous way as do the real rocks, but it is clearly not applicable to the third rock type (Bowland shale), for which the evolution of pore geometry and microstructure with pressure is evidently more complex.

4. Thanks for pointing out evident lack of clarity in the use of Terzaghi effective pressure. We have tidied up inconsistencies in the use of this term in the text.

5. Concerning the estimation of the pore pressure multiplier $n$ in the effective pressure law describing permeability, this was done for example by fitting data relating pore pressure $P_p$, total confining pressure $P_c$ and log permeability $k$ to a description of the form $\log k = a + b\, P_c + cP_p$ by linear or non-linear multiple regression, or by using an alternative empirical description as appropriate. The data can be condensed onto a single line by plotting $\log k$ vs effective pressure as $b(P_c + cP_p/b\,)$, in which case $n = c/b$. A note of clarification has been added to the text.

6. Concerning line 398, the comment presumably refers to $P_{eff} = P_c\,(1 - m\,P_p)$. This is correct, but the pore pressure coefficient $m$ refers to this effective pressure being the value that describes the effect of pore pressure on *elastic distortions* of the rock and its pore spaces, as defined by Nur and Byerlee (1971), not to the pore pressure coefficient $n$, which pertains to the effect of pore pressure on permeability.

7. Concerning line 489, Missing porosity; not copied over from eq. 12. Well spotted. As pointed out, for the purpose of this illustration, permeabilities to water and gas were assumed to be the same. However, permeability to water in clay bearing foliated rocks can be about 1/10 that of gas (Faulkner and Rutter 2001, op.cit.). This is smaller than the effect of large variations in effective pressure, but the point has now been raised in the text.

8. Error in denominators well spotted, now corrected. Concerning dimensions of $Nc^2d$ in eq.16 and the one before, $N$ has dimensions of m$^{-3}$, because it is the number of tubes of 1 m length in a 1 m$^3$ volume, as defined in the line following eq. 13. Thus $N$ is also the number of tubes intersecting a 1 m$^2$ cross section. Thus there is no dimensions conflict.

**Technical corrections (5):**

1. Line 63, Good point, 2nd decimal place on volume proportions removed.
2. Error bars larger than mean point, yes it is ok
3. Density of quartz entered wrongly by a factor of $\times 10$ ! Well spotted.
4. Line 181, rephrased.
5. Line 536, Well spotted.

**Additional comment posted by Prof . Christian David on 6 Jan 2022:**

Thanks for addressing my points.

My point number 5 was not correctly understood, I wanted to say that the effective pressure should be defined as Peff=Pc-mPp (like in line 256) and not Peff=Pc(1-mPp).

*Author's response on 6 Jan 2022:*

Thankyou for pointing out the typographic error. We did indeed misinterpret your question. You are quite right and your persistence is much appreciated.

Places in the marked-up version of the ms where the above changes have been applied are highlighted in yellow.